# Estimation of Doppler Velocity from Incoherent Scatter Spectra Using Context-Aware Transformers

Yanlin Li and Qihou Zhou
Department of Electrical and Computer Engineering
Miami University, OH 45056
Correspond to Qihou Zhou at zhoug@miamiOH.edu




#### Abstract

We present a context-aware transformer model for estimating Doppler velocity from incoherent scatter radar (ISR) spectra. The model is based on the standard transformer encoder with adaptations from the Vision Transformer. Trained entirely on theoretical spectra, the AI model generalizes well for Arecibo ISR data and outperforms the traditional fitting methods significantly. Simulations show that the velocity error of the conventional least-squares fitting (LSF) is 1.5 to 3.5 times that of the AI model using 5 input heights. An inference from the AI model is approximately 100 times faster than the LSF method and requires minimal hardware, making it practical for large-scale or real-time processing. The AI approach applies to all situations where the spectrum can be parameterized.

Keywords: Context-aware transformer; incoherent scatter radar; Doppler velocity estimation; Al optimization




#### 1. Introduction

Measuring the Doppler velocity of a medium using the power spectrum is a common problem in many applications. Incoherent scatter radars (ISR) provide one of the most direct and reliable means of measuring ionospheric parameters, including Doppler velocity, particularly during disturbed conditions. This study focuses on accurately determining the ionosphere velocity from the power spectra of ISR. Accurate Doppler velocity measurements are important for understanding ionosphere dynamics, monitoring geomagnetic activity, and improving space weather forecasts.

Doppler velocity from the power spectrum is traditionally derived using three main approaches: the moment, autocorrelation function (ACF), and the least-squares fitting (LSF) method (Woodman, R. F., 1983; Woodman & Hagfors, 1969; Li & Zhou, 2024). The moment method calculates the first moment of the Doppler power spectrum, yielding a weighted average velocity. The ACF method computes the ratio of the ACF's imaginary to the real part at different lags. The ACF and moment methods require only the power spectrum to be symmetric, but do not need any other knowledge of the power spectrum. Their easy implementations and computational efficiency make them a popular first choice. Nevertheless, the ACF and moment methods can be sensitive to noise and may not always have the desired accuracy. The least-squares fitting (LSF) method compares the measured power spectrum to theoretical spectra and estimates the Doppler shift and spectral width by typically minimizing the least-squares error. This approach is more accurate but computationally more demanding.

Recent advances in machine learning have seen the method used in diverse fields. Unlike
traditional methods, our results show that machine learning models learn directly from data and surpass traditional approaches, especially under noisy or complex settings. Transformer architectures, in particular, have shown strong results in a range of tasks due to their ability to extract relevant patterns from sequences using self-attention (Vaswani et al. 2017). Although originally developed for natural language processing, they have been adapted to structured inputs. We demonstrate here that transformers can process ISR spectral data and estimate Doppler velocity using context across the full input profile.

In the following two sections, we first describe the ISR data and then the AI model used for this work. In Section 4, we compare the AI results with the traditional LSF method using data taken by the Arecibo ISR data to demonstrate the former's advantages.

#### 2. Experimental and training data

All training and evaluation data are synthetically generated using the standard theoretical incoherent scatter spectrum model (Swartz & Farley, 1979; Kudeki & Milla, 2011). Each sample consists of 5 consecutive altitude bins, spaced 300 meters apart, with one incoherent scatter spectrum per altitude bin. The spectrum at each altitude is sampled at 101 points between ±12.2 kHz, with a resolution of 244.3 Hz and normalized to have a maximum value of 1. The bandwidth and frequency resolution are selected based on the maximum compatibility with the


existing Arecibo Observatory ISR data processing workflow described in Li & Zhou (2024). These hyperparameters can be easily modified to support different coding configurations or facilities.

In a typical Arecibo Coded Long Pulse (CLP) configuration with a 2 µs gate width, the full bandwidth is 500 kHz, corresponding to a Doppler aliasing limit of ±87.2 km/s. Typical line-of-sight ion velocities are below ±100 m/s, corresponding to a Doppler shift of about 287 Hz. The raw spectrum is computed from CLP data with a native resolution of 2.27 kHz and is interpolated to 244.3 Hz using FFT zero padding. The interpolation was originally introduced for compatibility with the traditional curve fitting method and is retained in this work without modification.

Although further interpolating the spectrum to a finer frequency grid may appear to be beneficial, we observe no performance gain once the network is sufficiently trained. The input head consists of a multi-layer perceptron (MLP) (Hornik et al. 1989) that processes the spectrum before it enters the transformer blocks. This learned projection serves as a data-driven alternative to fixed interpolation and is likely able to extract sub-bin Doppler information by
 learning smooth spectral structures directly from the input. Because the MLP operates across all frequency bins simultaneously, it can learn to resolve fine-grained shifts and spectral shapes without relying on increased frequency resolution.

In the synthesized training data, the Doppler velocity is randomly assigned for each sample, drawn uniformly from -100 to 100 m/s. The signal-to-noise ratio (SNR) is also randomly assigned, following a logarithmic distribution between 5 and 50 dB, representing the range from low-quality to near noise-free ISR measurements. All other plasma parameters, including electron density (N<sub>e</sub>), electron temperature (T<sub>e</sub>), ion temperature (T<sub>i</sub>), and the ion fractions of H<sup>+</sup>, He<sup>+</sup>, O<sup>+</sup>, and O<sub>2</sub><sup>+</sup>, are randomly sampled from real ISR measurements obtained through traditional LSF methods as discussed in Li and Zhou (2024, 2025a).

To generate a full vertical profile for each parameter, including SNR, a smooth nonlinear curve is constructed using the expression

$$y(i) = X_0 + (X_1 - X_0) \left(\frac{i}{N}\right)^{\alpha}, \quad i = 1, 2, ..., N$$
 (1)

where N is the number of consecutive altitude bins,  $X_0$  and  $X_1$  are the lower and upper bounds of the parameter value, centered around a given input value with a random range of variation up to 10%. For this study, we choose N=5 for our context-aware model and N=1 is context-unaware. The exponent  $\alpha$  is randomly selected from either the concave down range [1, 1.1], which produces a gently decreasing slope, or the concave up range [0.9, 1.0], which produces a curve that rises more steeply at lower altitudes and flattens at the top. Each curve is flipped in order with 50% probability to allow both increasing and decreasing trends.

An offset is used across the N heights to ensure the value at index integer(N/2)+1 equals the originally sampled target value. A total of 2 million training samples are generated using this process. An independent test set of 100,000 samples is constructed using the same procedure.

This approach aligns with broader definitions of physics-informed machine learning, where domain knowledge shapes the training data rather than being hard-coded into the model itself.

The measurements used for validation and comparison are data taken at the Arecibo Observatory at September 14, 2014, using the CLP data. The characteristics of the CLP program and the Arecibo instruments can be found in Sulzer (1986), Isham et al. (2000), and Li and Zhou (2024).

#### 3. Methods

# 100 3.1. Al architecture




We follow the standard transformer encoder architecture introduced by Vaswani et al. (2017), with adaptations based on the Vision Transformer (ViT) framework of Dosovitskiy et al. (2020). For brevity, we refer readers to these original works for detailed descriptions of the core architecture and attention mechanisms. Building on this foundation, we developed a context-aware deep learning workflow to estimate Doppler velocity directly from ISR-derived altitude profiles.

The input consists of spectral measurements across multiple altitudes, originally structured as a grid of 101 frequency points by 5 heights. The 5×101 input (heights × frequency) is first flattened into a 505×1 vector and passed through a Conv1D layer with stride 101 and output dimension 512, producing a 5×512 tensor. This step serves a dual purpose, as it restructures the data into 5 tokens with 512-dimensional embeddings compatible with the later self-attention layers, and it allows the convolutional filters to capture potential spectral correlations across adjacent height levels

Transformers do not have any built-in notion of token order or spatial structure, so positional encodings are required to provide this information. We use trainable positional encodings learned from data, which allow the model to capture patterns in structured inputs more effectively than fixed alternatives. This positional information is important for learning spatial dependencies relevant to Doppler velocity estimation.

Each transformer block consists of a multi-head self-attention mechanism followed by a feedforward network, with residual connections and layer normalization applied at each sublayer. We adopt a pre-normalization configuration, where layer normalization is applied before both the attention and feed-forward modules to improve training stability in deeper networks.

A dedicated trainable classification token, commonly referred to as [CLS] token in AI literature, is prepended to the input sequence and serves as a summary representation. In transformer architectures such as BERT or ViT (Devlin et al. 2019; Dosovitskiy et al. 2020). The [CLS] token interacts with all tokens via self-attention and is used as the final input to the regression head. We also evaluate an alternative strategy using global average pooling across all token outputs, as discussed in Section 3.2. An overview of the model architecture is shown in **Figure 1**.

Figure 1. Model overview. The architecture consists of 31 transformer encoder blocks, 321 layers in total, and approximately 100 million parameters.

### 3.2 training




The model is trained using 2 million synthetic samples and validated on 10,000 synthetic validation samples. The batch size is 512. Training is run for 30 epochs using the Adam optimizer and mean squared error (MSE) as the loss function. A cosine learning rate schedule is used with linear warmup for the first 1000 iterations. The learning rate starts at  $10^{-7}$  and linearly increases to  $10^{-4}$  after 1000 iterations. Then, the LR decays following a cosine curve, reaching  $10^{-7}$  of the initial learning rate by the final epoch.

Layer-wise Learning Rate Decay (LLRD) is a fine-tuning strategy commonly used in natural language processing, particularly for models like Bidirectional Encoder Representations from Transformers (BERT). We found that LLRD is necessary for stable training in our applications, particularly in deeper models. Without LLRD, deeper models fail to benefit from scaling. In many cases, increasing the number of transformer blocks leads to worse performance than smaller models. While our model can operate with as few as 1 transformer block, we observed consistent performance gain up to 31 blocks based on simulation results. Therefore, we adopt a 31-block architecture and apply LLRD with a decay rate of 0.9, which stabilizes training and enables effective scaling.

#### 3.3 Simulations and comparisons

We evaluate two key design choices in the model architecture: whether to use the full altituderesolved ISR input or an averaged spectrum, and whether to aggregate token representations






using a [CLS] token or global average pooling. The context-aware version (5ht-aware) treats each of the 5 altitude levels as a separate input token, preserving vertical structure and allowing the transformer to model inter-altitude dependencies through self-attention. The context-unaware version (context-unaware) averages the 5 spectra into a single profile, removing altitude information.

For aggregation, we compare global average pooling to a trainable [CLS] token. In the pooling variant, token outputs from the final transformer layer are averaged before being passed to the regression head. In the [CLS] configuration, a trainable token is prepended to the sequence and extracted after the final layer, allowing the model to learn a global representation directly from the full token set.

We first compare the two aggregation methods using the context-aware input. Once the better aggregation strategy is determined, we fix it and evaluate the impact of vertical context by comparing the context-aware and context-unaware variants. Finally, the traditional curve fitting method is included as a reference for comparison against the best-performing deep learning model.

#### 3.3.1 CLS vs global pooling

The two aggregation strategies differ in how the final representation is derived and fed to the output MLP. In the [CLS] configuration, a trainable [CLS] token is prepended to the input sequence before positional encoding. After passing through the transformer layers, only the final state of the [CLS] token is used as input to the output MLP, which produces the Doppler velocity prediction. In the global average pooling variant, no [CLS] token is used. Instead, the outputs of all tokens from the final transformer layer are averaged along the sequence dimension, and this pooled vector is passed to the output MLP. Both configurations use the same output head architecture, but differ in how information from the sequence is aggregated.

Our experiments show that the [CLS] aggregation strategy consistently outperforms global average pooling in terms of RMSE and scalability. With a shallow 2-block model, [CLS] achieves about 2 percent lower RMSE than global pooling. As the model scales to 31 blocks, the gap widens to roughly 5 percent. In contrast, global pooling does not benefit from increased depth, as deeper models show no performance gain and often exhibit unstable training behavior. Although global pooling may occasionally match the [CLS] model on specific validation runs, its overall performance is less stable. These findings indicate that global pooling is less effective in our setting and that the [CLS] token provides more robust and scalable performance.

## 3.3.2 context awareness

In traditional ISR spectral fitting, range integration or vertical smoothing is often applied before parameter estimation. This reduces noise by averaging incoherent scatter spectra across








altitude, but removes vertical structure. The context-unaware model adopts the same approach by averaging the 5 × 101 input across altitude into a single 101-point spectrum, treated as one altitude. Since self-attention requires multiple tokens, the 101-point spectrum is reshaped into 101 tokens with one feature each so that attention operates along the spectral dimension.

The context-aware model retains the full vertical structure by treating each altitude as a separate token. It takes all 5 incoherent scatter spectra directly, with each token representing one altitude and containing a 101-point spectrum. The transformer receives all 5 tokens and returns a single Doppler velocity prediction. The middle altitude bin (3<sup>rd</sup> height) is used as the prediction target.

Both models use the same 31-block [CLS]-based architecture but are trained and tested with different input formats. The context-aware model is trained on the full  $5 \times 101$  input. The context-unaware model is trained on standalone 101-point spectra with artificial noise and no vertical variation. In other words, it receives the average of the  $5 \times 101$  input, reshaped to  $101 \times 1$ .

To benchmark model performance, we compare the AI models against two LSF baselines using simulated data. The first scenario, LSF-ideal, assumes access to the true (noise-free) spectrum with known amplitude, which is not achievable in real measurements. The second, LSF-realistic, follows the approach discussed in Li & Zhou (2024, 2025), where plasma parameters, including Doppler velocity, are estimated from noisy spectra through parameterized fitting. Both LSF methods use the averaged spectrum from five heights to match the resolution of the AI models.

Figure 2(a) shows the root mean squared error (RMSE) as a function of the noise standard deviation ( $\eta$ ) and equivalent spectral bandwidth for the 5ht-aware model. Figure 2(b-d) show the RMSE ratios of the 5ht-aware model to the other three methods. The equivalent bandwidth characterizes the effective spectral width of the incoherent scatter spectrum and reflects the combined influence of ion temperature, mass, and composition (Zhou, 2002). Its range in Figure 2 spans the 430 MHz incoherent scatter spectral bandwidth from the E-region to the topside. In the simulations, the ground truth Doppler velocities follow a uniform distribution in the range between -85 to 85 m/s. The velocity RMSE from the 5ht-aware model in Figure 2(a) increases with  $\eta$  as expected. When  $\eta$  is above 30 (~10<sup>1.5</sup>), it is largely independent of the equivalent bandwidth.

The context-aware model consistently achieves lower RMSE than the LSF-realistic and context-unaware models across the full practical range of  $\eta$  values and equivalent bandwidths. While arithmetic averaging is most effective in reducing uncorrelated stationary Gaussian noise, the context-aware model implicitly functions as a denoising network. It has prior knowledge of the typical spectral shapes at different heights and learns to extract consistent features across the noisy inputs. As a result, it may suppress noise more effectively than simple arithmetic averaging and hence outperforms the context-unaware model. The LSF-ideal method outperforms the 5ht-aware model in the low noise regime, where the input spectrum is nearly noise-free and the fitting problem is well-conditioned. In this case, the spectrum is effectively a clean copy of the known target, and the algorithm can retrieve the Doppler largely without error. The RMSE of the





LSF-realistic method is about 1.5 and 3.5 times that of the 5ht-aware model for  $\eta$  at 0.1 and 0.01, respectively.

**Figure 2.** (a) RMSE in log<sub>10</sub>(m/s) as a function of noise standard deviation and equivalent bandwidth for 5ht-aware model. RMSE ratio of context-unaware (b), LSF-ideal (c) and LSF-realistic (d) to 5ht-aware model.

In **Figure 3**, we compare the performances of the 5ht-aware, LSF-realistic and the frequently used moment method as a function of altitude for a representative condition at Arecibo. Here we consider not only the noise standard deviation as in Figure 2, but also the bias as well. The velocity bias and standard deviation ( $\sigma$ ) are obtained from 24,795 runs with the same input velocity and noise standard deviation,  $\eta$ . The velocity is made to change with altitude as  $v(z) = A(z) \cos(\frac{2\pi}{(z-60)^{0.8}}(z-90))$ , where  $A(z) = 50(1-e^{-\frac{z-90}{10}})$  and z is the altitude in km.

v(z)/30 is depicted in Figure 3(a) as a dotted magenta line. The other three lines in Figure 3(a) are the biases, defined as the input velocity minus the results from the three methods. The ionosphere parameters and  $\eta$  are taken from representative daytime measurements on Apr. 12, 2013 at Arecibo. The 5ht-aware model has a comparable bias to the LSF method. The bias of LSF-realistic is approximately 3% of the true velocity for the noise standard deviation used. The LSF and moment methods underestimate the true velocity for the same reason that the mean velocity tends to zero in the absence of noise. It is of interest to note that the largest biases of the 5ht-aware model occur at the middle of the velocity range, likely due to the model's effort to compensate for the larger bias typically associated with higher velocities. LSF's standard deviation ( $\sigma_{LSF}$ ) does not only depend on  $\eta$  but also on the velocity amplitude.  $\sigma_{Moment}$  is linearly proportional to  $\eta$  for all the altitudes.  $\sigma_{AI}$  is the smallest among the three methods. To quantitatively show the improvement of the AI over the other two methods, we plot the ratio of velocity standard deviations in Figure 3(c). Averaging over 87 to 193 km,  $\sigma_{AI}$  is about 64% and 38% of  $\sigma_{LSF}$  and  $\sigma_{Moment}$ , respectively.

Figure 3. (a) Simulated biases and (b) standard deviations of the AI 5ht-aware model, LSF-realistic, and moment methods as a function of altitude. The magenta curve in the middle panel represents  $1000\eta$ . (c) Ratios of standard deviation of AI to LSF and moment methods.

# 4. Application to Arecibo ISR data processing

We apply the analysis technique to the data taken at Arecibo on July 16, 2015. During the period, the Arecibo linefeed rotated back and forth in the azimuth direction at a slew rate of 24°/min with a constant zenith angle of 15°. The raw data were processed to mitigate the interferences as discussed in Zhou et al. (2024) before computing the spectra. Figure 4(a) and 4(b) show the line-of-sight velocities from the context-aware model and the LSF method
discussed above. The integration time for the power spectrum is 30 sec. The setup is the same as in the above section, i.e., the power spectra are integrated over 5 heights to have a range resolution of 1.5 km, and the number of aware heights in the Al context-aware model is 5. As seen in the above section, the context-unaware and moment methods are inferior to the 5ht-aware and LSF-realistic method, respectively, and will not be discussed in this section.





Figure 4. (a) Line-of-sight velocity derived from the 5ht-aware model (upper panel), (b) from the LSF-realistic method (lower panel). Positive velocity is away from the radar.

The line-of-sight velocity,  $V_r$ , is a superposition of the horizontal and vertical velocities. The vertical stripes in the velocity plots are due to the constant rotation of the antenna. As we do not expect  $V_r$  to change randomly, its height coherence reflects the data quality. As seen from Figure 4, the AI plot shows better coherence than the LSF plot in the bottom. This is more clearly seen between 90 to 100 km and between 120 and 125 km. The amplitude in the LSF plot is smaller, as discussed above.

For a slowly varying quantity, the standard deviation of the second-order difference of independent samples is  $\sqrt{6}$  times of the random error, as measured by the standard deviation. **Figure 5(a)** shows statistical errors (divided by 40) of the 5ht-aware model (blue dots) and the LSF method (red dots) when the  $2^{nd}$  order partial difference is taken in the altitude direction. The error profiles are similar to that shown Figure 3(c), and the lowest error occurs at an altitude of 100 km. Both AI and LSF errors increase almost linearly from 100 km to about 180 km due to the increase in spectral width, and hardly vary below the F-region peak from 180 to 300 km. The average electron density profile for this period is plotted as a black line for background information. The average F-region peak altitude during this period is at 330 km. The error ratio of the 5ht-aware model to LSF-realistic method,  $\gamma$ , is plotted as a green line. In Figure 5(a), where the error is based on the  $2^{nd}$  order difference in altitude,  $\gamma$  is largely a constant above 120 km at 0.55. Below 100 km,  $\gamma$  is about 0.4. The about 50% error reduction in the AI model in Figure 5(a) is largely consistent with the results shown in Figure 3(c) and Figure 2(d).





We can also estimate the error by taking the 2<sup>nd</sup> order partial difference with respect to time. The results are shown in Figure 5(b). The AI error is much larger than that in Figure 5(a) even though the error trends remain the same while the LSF error is not much affected. A possible explanation for the difference in the error behaviors in Figure 5(a) and 5(b) is that the noise baseline, which needs to be subtracted from the spectrum before Doppler processing, is a function of frequency as well as time. How the noise baseline is estimated affects the results. It impacts the LSF method less because the fitting error is already large due to statistical fluctuation. In any event, the AI error is still 30% smaller than the LSF method around 110 km, which is the focus of the current study. Above 120 km, a larger number of heights can be used in the context-aware model to reduce the error.

Figure 5. (a) Average velocity errors (divided by 20) estimated from the 2<sup>nd</sup> order difference in altitude using the AI 5ht-aware model and the LSF-realistic method over the period of 06:30-15:25 LT on July 16, 2015. The green line is the ratio of AI to LSF error. The black line is the average electron density. (b) Same as (a) except that the 2<sup>nd</sup> order difference is taken in the time direction.

### 5. Conclusion

In this study, the AI context-aware model uses 5 heights to allow a good height resolution (1.5 km) for the E-region. At altitude ranges where coarser height resolution is acceptable, the number of heights in the context-aware model can be increased. This further elevates the advantage of the AI method. For example, the error ratio of the LSF-realistic to the AI context-aware model using 9 input heights is larger than 2 in practically all scenarios. Beyond accuracy, AI transformer models offer a computational advantage as well over the least-squares fitting method. Velocity inference is roughly 100 times faster than the fitting method and requires significantly fewer computational resources. Once trained, the model runs efficiently on modest hardware, making it suitable for real-time applications and large-scale data processing.

To conclude, we have developed a context-aware transformer model to determine the Doppler velocity from incoherent scatter spectra. Simulations and applications to the Arecibo incoherent scatter radar data show that the AI model consistently outperforms the traditional least-squares fitting method across a wide range of conditions, demonstrating strong generalization despite being trained entirely on synthetic data. Because the training data is based on physics-based simulations, the model is not limited to any specific radar and can be applied more broadly to any situation where the spectra can be parameterized.

# 325 Acknowledgments

The study is supported by NSF grants AGS-2152109 and AGS-2514168.

#### **Open Research**

The Arecibo raw data can be downloaded from the Texas Advanced Computing Center

(https://tacc.utexas.edu/research/tacc-research/arecibo-observatory/). The analyzed data discussed in this article are available in Li & Zhou (2025b).

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
