# Peer review of "Estimation of Doppler Velocity from Incoherent Scatter Spectra Using Context-Aware Transformers"

_EGUsphere, 2025_

## Referee Comment (RC2)

**Referee report on: "Context-Aware Deep Learning for Doppler Velocity Estimation from ISR Spectra"**

The manuscript proposes a transformer-based approach for estimating Doppler velocity from ISR spectra, trained entirely on synthetic data generated from standard ISR theory. The idea is interesting, but several scientific and methodological issues weaken the validity of the results. The current version does not show sufficient evidence that the method generalizes beyond the highly idealized training configuration, nor does it justify the architectural design or provide the necessary details for reproducibility.

The current version does not meet the level of rigor required for publication. Major revisions are needed. Until these issues are resolved, the reliability of the scientific conclusions remains uncertain.

Below I summarize the major concerns:

1. Oversimplified synthetic data

The model is trained exclusively on synthetic data generated under very restrictive assumptions. Equation (1) produces only smooth and monotonic profiles with limited variability. This excludes many structures that commonly appear in real ISR observations, such as sharp gradients, inversion layers, and localized irregularities.

Since the network never sees more realistic patterns during training, it is not possible to conclude that it will behave reliably when applied to real conditions. In addition, the evaluation metric used in the real-data analysis (second differences in height and time) strongly favors smooth profiles. A model trained on artificially smoothed data will naturally perform well under such a metric, even if the estimated Doppler velocities are biased or physically incorrect.

The manuscript does not discuss these limitations, and no experiments are presented to test the robustness of the method to deviations from the synthetic assumptions. Therefore, the reported improvements cannot be interpreted as evidence that the transformer is learning meaningful Doppler information. Instead, they appear to reflect the smoothness that is already imposed by the synthetic data.

The manuscript also refers to other works for parameter bounds, but these bounds should be explicitly stated here. The reader does not know the limits used to generate the synthetic data.

2. Extremely large transformer without justification

The proposed architecture is a 31-layer transformer with approximately 100 million parameters. This is extremely large considering that the input consists of only five altitude bins and 101 frequency channels. The manuscript does not justify why such a large model is necessary. The vertical variability is limited and the spectral dimension is fixed, under these circumstances simpler architectures (e.g., shallow transformers, small CNNs, or MLPs) could probably achieve similar performance.

The manuscript does not include comparisons with simpler models and does not analyze model efficiency. As a result, it is only demonstrated that "a very large transformer can work," not that this architecture is appropriate nor optimal. Without justification, the use of such a large model raises concerns that it may overfit the limited synthetic data distribution rather than learn robust physical behavior.

3. Incorrect characterization as physics-informed ML

The manuscript states that the approach "aligns with broader definitions of physics-informed machine learning." This is not accurate. A physics-informed model usually incorporates physical constraints directly into the architecture or the loss function (for example, PDEs or conservation laws). In this work, the model is a conventional transformer without any embedded physical constraint. The physics appears only through the synthetic spectra used for training. This distinction is important to avoid giving the impression that the network structurally incorporates ISR physics.

4. Unclear definition of network input and output

The manuscript does not clearly describe the exact input and output of the proposed neural network. It is not explicitly stated how the 5 altitude bins and 101 frequency channels are arranged, normalized, or preprocessed before being passed into the Conv1D layer. Likewise, it remains unclear whether the network outputs a single Doppler velocity for the entire 5-bin block or one velocity per altitude bin. This information is essential to understand the model, to evaluate the fairness of comparisons with LSF, and to reproduce the results. The authors should provide a precise description of the input tensor shape, preprocessing steps, tokenization, and the definition of the output quantity.

5. Insufficient description of the architecture

Several essential architectural details are missing. The number of attention heads, dropout usage, kernel sizes, feedforward dimensions, activation functions, normalization layers, positional-encoding formulation, and parameter initialization are not specified. For a model of this size, these details have a significant impact on performance and reproducibility.

In addition, the processing of the ISR spectra before they are given to the network, how the spectra are grouped, normalized, interpolated, and reshaped into tokens, is described only partially. These omissions prevent independent replication and validation.

6. Incomplete simulation and comparison methodology

Important elements of the training and evaluation pipeline are not well described: the noise model, sampling of SNR and bandwidth, instrumental effects, real-data preprocessing, and the exact LSF configuration. The real-data comparison relies only on smoothness-based metrics with no independent reference, making it difficult to assess true accuracy.

7. Limited applicability to other ISR systems

The manuscript suggests that the method could be applied to any ISR. However, the current model is strongly tied to the specific configuration of the Arecibo 430 MHz system. The synthetic spectra used for training assume a fixed radar frequency, fixed altitude resolution, fixed frequency resolution, fixed bandwidth, and a specific SNR and spectral shape that correspond only to this station.

Other ISR systems (e.g., EISCAT, Millstone Hill, Jicamarca) use very different pulse codes, ambiguity functions, altitude sampling, integration schemes, and spectral characteristics. Because the model learns the distribution of spectra produced by the Arecibo configuration, it cannot be directly transferred to another system. Adapting the model would require generating a completely new synthetic dataset (including all the possible REAL profiles) consistent with the new radar and then retraining the model from the beginning.

For this reason, the claim of broad applicability is overstated, and the manuscript should clearly acknowledge that the method is not easily portable to other ISR facilities.

---

## Author Comment (AC1)

Response to Reviewer 1's comments

Reviewer's comments are in black. Responses are in blue.

The manuscript reports results of line-of-sight plasma velocity fits to Arecibo ISR data using artificial intelligence (AI), namely context-aware transformers. The manuscript seems to be continuation to a series of papers by the same authors [1,2,3], in which they apply different data analysis techniques to archived coded long pulse (CLP) data from the Arecibo radar. The results suggest that AI may produce high-quality results in ISR data analysis.

While the idea to replace the traditional least-square fitting techniques with computationally less expensive (after the expensive training has been done) AI techniques is novel and the results look promising in general, I have several critical comments about the text and interpretation of the results.

1. The manuscript lacks critical references and fails to explain key principles of the AI model. If the idea is to introduce the AI techniques to the ISR community, skipping most of the key information "for brevity" may not be the best choice. It is understandable that Section 3.1, which describes the AI architecture, is full of field-specific jargon, but the terminology should be explained to the reader in such a level that reading the text is possible also for a non-expert of the field without reading all the references, and references to the key concepts should be given for readers who are interested in more details.

Response: Section 3.1 has been significantly expanded to clarify field-specific terminology and to explain the terms of the transformer architecture in a manner accessible to non–machine-learning experts. We have also included a number of references in several areas.

2. The actual scientific target of the measurements considered remains unclear. The authors first give a few very general motivations for measuring the Doppler velocities (without references). The AI model, which solves only for plasma velocities, is then compared with a least-squares fitting technique, which fits also several other parameters. Is there some specific application, for which the velocities alone are important? Would it be possible to use the AI model to fit the same parameters that are fitted with the least-squares solver? On line 300 the authors finally claim that focus of

this study is around 110 km altitude. What exactly is the focus and why not to mention it in the abstract and in the introduction?

Response: The target of the manuscript is to estimate Doppler velocity from incoherent scatter spectra using context-aware transformers. Other than ISRs, Doppler velocity determination is a common functionality for most radars, such as weather and air-traffic control radars. The region around 110 km altitude has a larger vertical gradient in velocity and electron density. The rapid changes make it easier to contrast the differences between different techniques. We have added more references on Doppler measurements and a sentence in the abstract to indicate the focus around 110 km.

3. Least-square fits contain several tunable parameters that may greatly affect quality of the results, but these are not considered at all. In particular, stopping criteria for the iterations and initial values of the fitted parameters may affect both standard deviation and bias of the results. These should be carefully evaluated when comparisons between the AI model and the least-squares fits are performed.

Response: In the present study, the least-squares baseline does not involve a general iterative optimization with multiple tunable convergence parameters. Doppler velocity is estimated by minimizing the squared error between the measured and modeled spectra over a one-dimensional grid of Doppler shifts with a fixed resolution (0.1 m/s). For this formulation, the objective function admits a unique global minimum for a given resolution, and the resulting solution is therefore deterministic and independent of both initialization and stopping criteria. The following text has been added to the end of section 3.3.

'Finally, the traditional curve fitting method is included as a reference for comparison against the best-performing deep learning model, where Doppler velocity is obtained by a one-dimensional least-squares search over a discretized Doppler shift grid with a resolution of 0.1 m/s. For this formulation, the cost function has a unique global minimum at the chosen resolution, yielding a deterministic solution that is independent of initialization and stopping criteria. The fitting algorithm is described in Li & Zhou (2024).'

4. Computational requirements are not discussed at all until the Conclusions section, where the authors claim that "Velocity inference is roughly 100 times faster than the fitting method and requires significantly fewer computational resources.". This may be true, but some key figures about computational resources needed for both training the model and the final velocity inference should be given. Also the training part is

important for potential users of the technique, because it seems that one may need to train the model for each radar and radar operation mode separately.

Response: We have revised the manuscript to explicitly discuss the computational requirements of both training and inference.

The relevant text in the conclusion section is expanded to 'Beyond accuracy, the proposed transformer model also offers a clear computational advantage over the least-squares fitting method. Once trained, Doppler velocity inference is approximately 100 times faster than curve fitting and requires substantially fewer computational resources. The model contains approximately 100 million parameters (≈300 MB) and runs efficiently for inference on any modern discrete GPU, making it suitable for large-scale data processing and near–real-time applications. Model training is performed using synthetic data and can be completed in approximately two days on a higher-end GPU (e.g., RTX 3090 or above).'

5. The Arecibo radar collapsed a few years ago, but there are several other incoherent scatter radars in the world. Re-analysis of the archived Arecibo data is indeed valuable, but the authors could also comment if their technique might be usable for data from other radars that have considerably lower SNR and operate in completely different geophysical environments. In particular, other radars may observe much larger velocities and the users are typically interested also in electron densities and electron and ion temperatures, not just plasma velocities. At very end of the conclusions the authors claim, without any justification, that the model can be applied more broadly, but the very different noise levels and very much larger line-of-sight velocities observed with many other ISRs are not discussed at all.

Response: Although the examples in this study are designed specifically for Arecibo ISR, the model itself is trained entirely on synthetic ISR spectra, for which radar parameters, Doppler velocity range, and SNR are manually controlled. Therefore, the training set can be easily adapted to a different SNR and Doppler velocity range.

The relevant text has been revised to 'As the training data are generated using physics-based ISR simulations, SNR and Doppler velocity range can be explicitly controlled during data generation. Therefore, the proposed framework is not inherently limited to the Arecibo ISR and can be adapted to other instruments by retraining the model using instrument-specific parameters and configurations.'

Detailed comments:

Lines 22-23: "particularly during disturbed conditions."

Does this mean that the radars are more reliable than other instruments during disturbed conditions?

Response: Doppler measurement from radars is the most prevalent and recognized technique to measure the plasma drift velocity. The instrument's accuracy depends mostly on the level of ionization. We have removed the "disturbed conditions" in the revision.

Line 25-26: References to studies where these measurements are valuable would be useful.

Response:  More references are added.

Line 28: To my understanding, the moment and autocorrelation methods are not commonly used for ISR data analysis, because computers are powerful enough for the least-squares fits and the users are typically interested in many other plasma parameters as well. Please correct me if I am wrong.

Response: The Arecibo velocity data in the CEDAR-Madrigal database are based on moment or autocorrelation methods. We do not know exactly how other ISR sites derive the line-of-sight velocities, but suspect that this may still be the prevailing method. Obtaining the velocity separately is typically not about computational load. It is more about reducing the number of free parameters in the least-squares fitting to improve the accuracy and convergence of other parameters. We have added " More importantly, fitting the Doppler shift along with other ionosphere parameters makes the LSF less accurate and more difficult to converge on the optimal solution" at the end of the paragraph.

Lines 33-34: "Their easy implementations and computational efficiency make them a popular first choice."
Again, is this still true for IS radars nowadays?

Response: Please see the response above.

Lines 39-40: "Unlike traditional methods,..."
Does this refer to some traditional machine learning methods, or to the traditional radar data analysis methods?

Response: By "Traditional", we meant non-machine learning methods. We have changed the phrase to "Unlike fitting methods".

Line 59: Please give a reference to the coded long pulse technique.

Response: We have moved the references for CLP from a later part to where CLP is first mentioned.

Line 64: "...with the traditional curve fitting method."
Please explain what is "the traditional curve fitting method", and give a reference.

Response: "The traditional curving fitting method" is the least-squares fitting method. We have changed the sentence to: "The interpolation was originally introduced for compatibility with the LSF method used in Li and Zhou (2024) and is retained in this work without modification."

Equation (1): Shape of this profile seems to affect the final results, because the context-aware AI model learns this profile shape. Is there some physical justification for the selected function?

Response: Equation (1) constrains the maximum vertical variation of plasma parameters over the 1.5 km altitude range, with hyperparameters selected empirically based on variability observed in real ISR measurements.

Lines 85-86: "context-aware" and "context-unaware" are here used without explaining the terms first.

Response: We added 'a context-aware model that incorporates information from adjacent altitude bins, and a context-unaware model that processes each altitude profile independently'. after the relevant texts.

Lines 93-94: Please give references to the "broader definitions".

Response: The wording referring to "broader definitions" has been removed in the revised manuscript.

Section 2: It would be useful to show some examples of the synthetic IS spectra with different noise levels.

Response: The following sentence has been added to the first paragraph of Section 2 to direct the reader to references where incoherent scatter and synthetic spectra are discussed extensively. One additional reference has also been included.

'Representative examples of synthetic incoherent scatter spectra at different noise levels can be found in Aponte et al. (2006) and Li & Zhou (2024).'

Sections 3.1, 3.2 and 3.3: Please explain the AI terminology so that also readers who are not familiar with it can follow the description at least superficially, and give sufficient references. I will not list every single point separately in these comments.

Response: Done

Lines 124-125: "In transformer architectures such as BERT or ViT (Devlin et al. 2019; Dosovitskiy et al. 2020)."
This sentence seems to be completely detached from the surrounding text.

Response: This was a typo in the original manuscript. It was supposed to be a comma rather than a period after the sentence.

Line 199: "...context-unaware model is trained on standalone 101-point spectra with artificial noise..."
Is this noise somehow different from the noise added to the 5x101 input of the context-aware model?

Response: No. The same noise variance is applied independently at each height in both models. The context-aware model uses all five height-resolved spectra as separate tokens, while the context-unaware model averages the five heights.

Lines 208-217 & Figure 2. I do not understand what LSF-ideal and LSF-realistic mean here and how the comparison is done. The contours in Figure 2 are as function of bandwidth and noise std, but then the authors claim that there was no added noise (noise std=0?) in the LSF-ideal case. Please explain what happens in the comparison.

Response: LSF-ideal and LSF-realistic differ only in how the spectrum used for Doppler fitting is obtained. In the LSF-ideal case, we assume the true (noise-free) spectrum shape is known and retrieve the Doppler velocity by shifting this fixed template along the frequency axis and minimizing the least-squares error. LSF-Ideal represents the limiting case for the LSF method. In the LSF-realistic case, the spectrum shape is unknown and must first be estimated from noisy data. To make it easier to read, we have also added the explanation in the Figure 2 caption.

Lines 234-235: "...frequently used moment method..."
Please provide references that demonstrate the frequent use of the moment method in ISR data analysis.

Response: References are added in the introduction. "frequently used" is removed. Please see the response to the "Line 28" comment.

Lines 243-244: Is the bias in the LSF results possibly affected by the initial parameter values? One might expect this kind of bias profile if the iteration starts from zero velocity and tends to stop a bit too early.

Response: No. The least-squares Doppler estimation has a unique optimal solution and is invariant to the initial parameter value; the bias is therefore not related to early stopping.

Lines 244-246: "The LSF and moment methods underestimate the true velocity for the same reason that the mean velocity tends to zero in the absence of noise."
I do not understand this sentence. What is "the same reason"?

Response: The original sentence has a typo. "in the absence of noise" should have been "in the absence of signal". The sentence is now changed to: "In the extreme case of all noise and no signal, the mean LSF and moment velocities tend to zero because the estimated velocities are symmetrically distributed at positive and negative values. Similarly, as long as there is noise, LSF and moment techniques tend to underestimate the velocity amplitude."

Lines 248-249: Does the LSF standard deviation depend also on stopping criteria of the iteration? If the criteria are too loose, the iteration might stop at random locations around the true minimum of the cost function, increasing the noise.

Response: No. The least-squares fitting used here does not rely on an iterative optimization with stopping criteria. For each profile, the cost function has a unique global minimum that is deterministically identified, so the standard deviation of the LSF results is not affected by stopping criteria.

Figure 3, panel c: please change the colors, especially yellow is almost invisible.

Response: Done

Lines 279-280: "For a slowly varying quantity, the standard deviation of the second-order difference of independent samples is √6 times of the random error, as measured by the standard deviation."
Please give a reference.

Response: We have expanded the explanation. The revised version reads: We use the standard deviation of the second-order difference to estimate the velocity error. The second order difference, y, of a signal x at time $t_i$ is $y(t_i) = x(t_i + \Delta t_i) - 2x(t_i) + x(t_i - \Delta t_i)$, where $\Delta t_i$ is the sampling interval. For a slowly varying signal with superposed noise, the variance of y is 6 times the variance of x. The standard deviation

of the second-order difference of independent samples is thus $\sqrt{6}$ times of the random error, as measured by the standard deviation.

Lines 294-296: Is it possible that the fluctuations are true temporal variations in the wind field?

Response: The fluctuations can be from the velocity field (even though the second-order derivative can remove the linear variation). We have added this possibility in the revision. The added text reads: "Another possibility is that the standard deviation also contains non-linear temporal variations in the velocity field."

Lines 299-300: "In any event, the AI error is still 30% smaller than the LSF method around 110 km, which is the focus of the current study"
What exactly is the focus of the current study, and why is this mentioned only on line 300?

Response: We have added the rational of focusing on ~110 km at the beginning of Section 4. The added text reads: "We will mainly focus on the comparison in the E-region around 110 km where the vertical velocity and electron density gradients are larger than in the F-region. The larger gradients limit the height range one can integrate, and the effect of different signal processing techniques can be more readily seen."

Caption of Figure 5: (divided by 20) -> (divided by 40)?

Response: Thanks for the catch. It should have been divided by 40.

Conclusions: The conclusions should summarize the results and they should preferrably be understandable without reading the whole manuscript. Neither of these conditions is fulfilled in this case. The contents of the first paragraph would better fit to the preceding sections, and the discussion about computing resources should be expanded there.

 Response: We have rewritten the conclusion.

References used in the response.

Aponte, N., M. P. Sulzer, M. J. Nicolls, R. Nikoukar, and S. A. Gonzalez, Molecular ion composition measurements in the F1 region at Arecibo." Journal of Geophysical Research: Space Physics 112.A6, 2007.

Li, Y., and Zhou, Q.: Measurements of F1-region ionosphere state variables at Arecibo through quasi height-independent exhaustive fittings of the incoherent scatter ion-line spectra, J. Geophys. Res. Space Phys., 129(11), e2024JA032620, 2024.

Li, Y., and Zhou, Q.: Accurate spectral fitting in the upper F-region using the randomly coded data of the Arecibo 430 MHz radar, J. Geophys. Res. Space Phys., 130, e2025JA033877, https://doi.org/10.1029/2025JA033877, 2025a.

Zhou, Q., Li, Y., and Gong, Y.: Variance estimations in the presence of intermittent interferences and their applications to incoherent scatter radar signal processing, Atmos. Meas. Tech., 17(14), 4197–4209, https://doi.org/10.5194/amt-17-4197-2024, 2024.

---

## Author Comment (AC2)

Response to Reviewer 2's comments

Reviewer's comments are in black. Responses are in blue.

The manuscript proposes a transformer-based approach for estimating Doppler velocity from ISR spectra, trained entirely on synthetic data generated from standard ISR theory. The idea is interesting, but several scientific and methodological issues weaken the validity of the results. The current version does not show sufficient evidence that the method generalizes beyond the highly idealized training configuration, nor does it justify the architectural design or provide the necessary details for reproducibility.

The current version does not meet the level of rigor required for publication. Major revisions are needed. Until these issues are resolved, the reliability of the scientific conclusions remains uncertain.

Response: We have modified the manuscript extensively and hope we have addressed the issues raised by the reviewer. As the method requires only the theoretical model and the observed spectra, it is applicable to all cases where the least-squares fitting technique is used. Training for the model is general and does not have any restrictions beyond the incoherent scatter theory.

Below I summarize the major concerns:

1. Oversimplified synthetic data

The model is trained exclusively on synthetic data generated under very restrictive assumptions. Equation (1) produces only smooth and monotonic profiles with limited variability. This excludes many structures that commonly appear in real ISR observations, such as sharp gradients, inversion layers, and localized irregularities.

Since the network never sees more realistic patterns during training, it is not possible to conclude that it will behave reliably when applied to real conditions. In addition, the evaluation metric used in the real-data analysis (second differences in height and time) strongly favors smooth profiles. A model trained on artificially smoothed data will naturally perform well under such a metric, even if the estimated Doppler velocities are biased or physically incorrect.

The manuscript does not discuss these limitations, and no experiments are presented to test the robustness of the method to deviations from the synthetic assumptions. Therefore, the reported improvements cannot be interpreted as evidence that the transformer is learning meaningful Doppler information. Instead, they appear to reflect the smoothness that is already imposed by the synthetic data.

The manuscript also refers to other works for parameter bounds, but these bounds should be explicitly stated here. The reader does not know the limits used to

generate the synthetic data.

Response: The synthetic spectra cover all practical combinations. In ISR applications, the synthesized data with noise added are observations without instrumental problems. The smooth profile assumption is within the 1.5 km height resolution. Any variation with a scale larger than 1.5 km is completely resolved. 1.5 km is a well-accepted fine resolution for ISR applications.

The instrument limit at Arecibo is 300 m, which is practically a limit for all ISRs. If the instrument-limited resolution of 300 m in the output is required, one must use the height-unaware model. While we see in Figure 2 that AI height-unaware model can do better than LSF, it does not offer the same advantage as the height-aware model if a reduced resolution is acceptable.

One way to think about Equation (1) is to compare it with traditional ways to do averages.  In traditional approaches, all the heights are averaged with a fixed set of weights (often equal weights). Equation (1) allows the weights to vary dynamically and thus makes optimization possible. We would like to point out that the most essential information needed for training the model is the theoretical spectra. While our tests show that Equation (1) works reasonably well for the purpose of context awareness, there can be other forms of the equation that may work even better.

In short, the AI model does not assume or require anything more than the LSF fitting technique.

2. Extremely large transformer without justification

The proposed architecture is a 31-layer transformer with approximately 100 million parameters. This is extremely large considering that the input consists of only five altitude bins and 101 frequency channels. The manuscript does not justify why such a large model is necessary. The vertical variability is limited and the spectral dimension is fixed, under these circumstances simpler architectures (e.g., shallow transformers, small CNNs, or MLPs) could probably achieve similar performance.

The manuscript does not include comparisons with simpler models and does not analyze model efficiency. As a result, it is only demonstrated that "a very large transformer can work," not that this architecture is appropriate nor optimal. Without justification, the use of such a large model raises concerns that it may overfit the limited synthetic data distribution rather than learn robust physical behavior.

Response: The task addressed here is not a generic regression from a small input, but the extraction of a weak Doppler signal embedded in spectrally structured noise.

While the proposed model contains approximately 100 million parameters, this scale is comparable to widely used base configurations of modern transformer architectures (e.g., ViT-Base) and is not unusually large by current standards.

During model development, we examined the effect of transformer depth on Doppler velocity estimation accuracy. As shown in the blue curve below, the blue curve (CLS variant) observes performance gain up to 100 transformer blocks and perhaps beyond. The 31-block architecture in the present work was selected to balance accuracy and computational cost.

[Figure]

3. Incorrect characterization as physics-informed ML

The manuscript states that the approach "aligns with broader definitions of physics-informed machine learning." This is not accurate. A physics-informed model usually incorporates physical constraints directly into the architecture or the loss function (for example, PDEs or conservation laws). In this work, the model is a conventional transformer without any embedded physical constraint. The physics appears only through the synthetic spectra used for training. This distinction is important to avoid giving the impression that the network structurally incorporates ISR physics.

Response: The statement referring to "broader definitions of physics-informed machine learning" has been removed in the revised manuscript. The proposed model does not incorporate physical constraints directly into the architecture or loss function; physics enters only through the synthetic training data.

4. Unclear definition of network input and output

The manuscript does not clearly describe the exact input and output of the proposed neural network. It is not explicitly stated how the 5 altitude bins and 101 frequency channels are arranged, normalized, or preprocessed before being passed into the Conv1D layer. Likewise, it remains unclear whether the network outputs a single

Doppler velocity for the entire 5-bin block or one velocity per altitude bin. This information is essential to understand the model, to evaluate the fairness of comparisons with LSF, and to reproduce the results. The authors should provide a precise description of the input tensor shape, preprocessing steps, tokenization, and the definition of the output quantity.

Response: The network input consists of ion-line spectra sampled at five adjacent altitude bins, each with 101 frequency channels (5×101). The network outputs a single Doppler velocity for the full 5-bin window. Using five adjacent heights provides vertical context, as opposed to the least-squares fitting baseline, which typically averages several heights into a single profile before estimating Doppler velocity.

5. Insufficient description of the architecture

Several essential architectural details are missing. The number of attention heads, dropout usage, kernel sizes, feedforward dimensions, activation functions, normalization layers, positional-encoding formulation, and parameter initialization are not specified. For a model of this size, these details have a significant impact on performance and reproducibility.

In addition, the processing of the ISR spectra before they are given to the network, how the spectra are grouped, normalized, interpolated, and reshaped into tokens, is described only partially. These omissions prevent independent replication and validation.

Response: The model description in the revised manuscript has been substantially expanded.

The model uses a standard transformer encoder architecture. Each block consists of multi-head self-attention with 4 heads (128 features per head) followed by a feed-forward network with width 4×d_model, consistent with the original transformer design. Layer normalization and residual connections follow the standard formulation. These details have now been explicitly stated in the revised manuscript.

6. Incomplete simulation and comparison methodology

Important elements of the training and evaluation pipeline are not well described: the noise model, sampling of SNR and bandwidth, instrumental effects, real-data preprocessing, and the exact LSF configuration. The real-data comparison relies only on smoothness-based metrics with no independent reference, making it difficult to assess true accuracy.

Response: The noise model, SNR and bandwidth sampling, instrumental effects, real-data preprocessing, and the exact LSF configuration are described in Section 2.

One may not necessarily agree on the methodology of error computation. What matters most in the current study is the comparison between the LSF and AI results. We use exactly the same metrics for AI and LSF.

7. Limited applicability to other ISR systems
The manuscript suggests that the method could be applied to any ISR. However, the current model is strongly tied to the specific configuration of the Arecibo 430 MHz system. The synthetic spectra used for training assume a fixed radar frequency, fixed altitude resolution, fixed frequency resolution, fixed bandwidth, and a specific SNR and spectral shape that correspond only to this station.

Other ISR systems (e.g., EISCAT, Millstone Hill, Jicamarca) use very different pulse codes, ambiguity functions, altitude sampling, integration schemes, and spectral characteristics. Because the model learns the distribution of spectra produced by the Arecibo configuration, it cannot be directly transferred to another system. Adapting the model would require generating a completely new synthetic dataset (including all the possible REAL profiles) consistent with the new radar and then retraining the model from the beginning.

For this reason, the claim of broad applicability is overstated, and the manuscript should clearly acknowledge that the method is not easily portable to other ISR facilities.

Response: The synthetic training data consist of generic ISR ion-line spectra and are not limited to a specific facility. Radar frequency, bandwidth, altitude resolution, and noise characteristics are explicitly configurable in the simulations, allowing the same framework to be adapted to other ISR systems by regenerating instrument-specific synthetic data and retraining the model. Such retraining is expected when adapting the model to other facilities and is straightforward. The main difference between Arecibo and other facilities is in the signal-to-noise ratio. Figure 2 gives a large range of normalized noise standard deviation that can be applied to other ISR facilities.